# Genome-Wide Association Study Reveals Novel QTNs and Candidate Genes Implicated in Resistance to Northern Corn Leaf Blight in Maize (*Zea mays* L.)

**DOI:** 10.3390/ijms262110677

**Published:** 2025-11-02

**Authors:** Udaya Shetty, Muntagodu Shreekanth Sowmya, Hirenallur Chandappa Lohithaswa, Mallana Goudra Mallikarjuna, Ganiga Jadesha, Siddaiah Chandra Nayaka

**Affiliations:** 1Molecular Plant Pathology Laboratory, Department of Studies in Biotechnology, University of Mysore, Mysore 570006, Karnataka, India; 2Department of Genetics and Plant Breeding, College of Agriculture, GKVK, University of Agricultural Sciences, Bangalore 560065, Karnataka, India; 3Division of Genetics, Indian Agricultural Research Institute, New Delhi 110012, India; 4Department of Plant Pathology, College of Agriculture, GKVK, University of Agricultural Sciences, Bangalore 560065, Karnataka, India

**Keywords:** maize, northern corn leaf blight (NCLB), resistance, GWAS, QTNs, candidate genes

## Abstract

Northern corn leaf blight is a major fungal disease hindering maize production worldwide. Among the various strategies of disease management, the deployment of host plant resistance is the most economic means to mitigate the yield losses, as it is cost-effective and durable. In this study, we performed the genome-wide association study (GWAS) analysis in a set of 336 maize inbred lines. The experimental material was evaluated for northern corn leaf blight disease response across two seasons during the rainy seasons of 2023 and 2024. The ANOVA results and estimates of genetic variability parameters indicated the existence of a substantial amount of genetic variability. High heritability and high genetic advance as percent mean suggested the presence of additive genetic effects in controlling the disease response. GWAS analysis was performed employing GLM, MLM, CMLM, MLMM, FarmCPU and BLINK. The results from GWAS identified 74 marker associations from GLM and FarmCPU models. The QTN *S1_7356398*, located on chromosome 1, identified from the GLM model, explained 12.12 percent of phenotypic variation. Another QTN *S2_51098833* located on chromosome 2, identified from the FarmCPU model, explained 6.14 percent variation. Remaining associations explained lesser PVE, suggesting the quantitative inheritance of NCLB resistance. Candidate gene identification was performed by keeping B73 as a reference genome. The identified QTNs from the current study were found to be located in annotated genes with functional domains implicated in defence mechanisms in maize and other crops. Many candidate genes, including chitinase, putative serine/threonine protein kinase, and aldehyde oxygenase, were identified and found to play a crucial role in plant defence mechanisms against several biotic and abiotic stresses.

## 1. Introduction

Northern corn leaf blight is a major foliar disease of maize, affecting maize cultivation and production in nearly all temperate and tropical maize-growing regions worldwide. It is caused by a hemibiotrophic pathogen, *Setosphaeria turcica* with anamorphic stage *Exserohilum turcicum* [Pass] Leonard and Suggs. It has a widespread existence throughout the globe and is commonly seen in Asia, America, Europe, and Africa. High humidity, low temperature, and cloudy weather are highly conducive for the disease development [1]. Based on the extent of disease severity, the yield loss can vary between 50% and 70% when infection occurs before silking and it affects the leaf area before seed set [2]. However, in India, yield loss can vary between 25 and 90%, depending on the disease severity. The premature death of the blighted leaves significantly reduces the active leaf area available for photosynthesis, causing severe yield losses.

Various races of *Exserohilum turcicum* exist in nature; the commonly occurring ones are 0, 1, 2, 3, 12, 23, 23 N, and 123 N, based on their virulence against host resistance genes viz., Ht1, Ht2, Ht3, Ht4, HtM, HtP, Htn1, HtNB, and rt in maize [3]. These Ht genes in maize confer qualitative resistance, and are race-specific, single-gene-inherited, and mostly dominant. However, the expression of these resistance genes in the host plant, or the avirulence genes in the pathogen, is greatly altered by environmental conditions, such as temperature, humidity, and light intensity, creating an unstable and less durable resistance. Many of the above-mentioned resistance genes in maize have been fine-mapped and cloned and are extensively employed in disease resistance breeding [4].

Although several qualitative Ht resistance genes have been cloned and pyramided in the breeding programmes, their race-specific nature and environmental instability limit their long-term effectiveness [4]. In the temperate environments where pathogen variability is comparatively low, pyramiding several Ht genes is considered to be an effective strategy in breeding NCLB resistance. In contrast, high pathogen abundance and variability in the tropical regions demand highly durable quantitative resistance (QTLs) to tackle the NCLB disease [5]. Small and additive effects of individual contributing alleles in a quantitative locus cannot be overcome easily by the evolution of pathogen races, and hence, it is more practically useful to the breeders. Several researchers reported the predominance of additive gene action controlling the resistance [6,7].

QTL or linkage mapping is an effective approach when studying the complex and polygenically inherited traits [4,8]. A number of mapping studies have been conducted to identify the genomic regions conferring NCLB resistance in various genetic backgrounds in different environments [6,7,9,10] Though it is a very powerful strategy, it has several limitations, such as a limited number of recombination, less allelic diversity (only two alleles per locus can be harnessed), and poor mapping resolution, as no strong inferences about linkage relationships among the identified QTLs can be made [11]. In most of the QTL mapping studies, the mapping populations and breeding populations are unrelated, and, hence, the translation of the QTLs identified to breeding targets has been very limited.

Genome-wide association studies (GWAS) offer a high-resolution alternative to QTL mapping by leveraging natural allelic diversity in diverse populations [12]. It involves the correlation of allelic frequencies at each of the several hundred thousand markers distributed throughout the genome with trait variations in a population-based sample, and such studies are based on the accurate phenotypic analysis of a given target trait in a large set of widely unrelated individuals. GWAS has made significant progress in the past decade and is widely employed to identify the allelic variants linked to various biotic and abiotic stresses in maize [13]. Several researchers employed GWAS to identify the genetic variants associated with NCLB resistance, [4,14,15,16,17] revealing the quantitative inheritance of NCLB resistance.

GWAS enables the detection of the genomic region associated with the trait of interest; however, pinpointing the causal candidate genes is necessary to translate these associations into functional insights. It is also essential to elucidate the genetic and molecular basis of host–pathogen interactions, thereby leading to a better understanding of resistance mechanisms. Previous QTL mapping and GWAS studies conducted across all ten chromosomes have identified several QTLs and associated candidate genes, including LRR-RLKs, PR proteins, P450s, transcription factors, and AAA-ATPase enzymes [15,18]. Bulk segregant analysis (BSA) and SNP-based linkage or GWAS studies have identified several QTLs associated with NCLB resistance in maize. Several candidate genes within these QTL regions have been reported, many of which are involved in key defence mechanisms, including ATP binding, kinase activity, and reactive oxygen species (ROS)-mediated signalling pathways [19,20]. Recent studies combining QTL mapping and GWAS in multi-parental populations have highlighted major genomic regions and candidate genes, such as pentatricopeptide repeat (PPR) proteins, which show an elevated expression in resistant lines under disease challenge, suggesting their potential role in conferring durable NCLB resistance [17]. Together, these studies build a picture of the complex, quantitative resistance of NCLB response in maize, mediated by many genes of moderate effect and involving multiple biochemical defence pathways. Although GWAS provides powerful insights, functional validation is essential to confirm the roles of identified loci in disease resistance.

We hypothesize that the natural variation among the maize inbreds contributes to differential resistance to NCLB, and that GWAS can identify genomic regions and candidate genes associated with it. In this light, the present study aims to perform GWAS analysis to identify QTNs linked to NCLB resistance and to pinpoint the candidate genes, located within the associated genomic regions, conferring NCLB tolerance, for future functional validation and use in resistance breeding programmes.

## 2. Results

The significance of the mean sum of squares due to the genotypes indicated the presence of substantial genetic differences among the genotypes, leading to a varied phenotypic response for NCLB disease (Table 1). After confirming the homogeneity of error variance through Levene’s test, a pooled ANOVA was performed, and its results are presented in Table 2. The high significance of the mean sum of squares, due to genotypes, years, and genotype × year interaction suggested the existence of genetic variability, along with the influence of environment on the disease expression. Furthermore, the genetic variability parameters of phenotypic and genotypic coefficients of variation, genetic advance as percent mean, and broad-sense heritability were estimated (Figure 1). The higher magnitude of the phenotypic coefficient of variation (PCV), genotypic coefficient of variation (GCV), heritability, and genetic advance as percent mean indicated the presence of additive gene action in controlling the disease response. Of the 336 maize inbreds, 8 were resistant, 241 were moderately resistant, 79 were moderately susceptible, and 8 were susceptible (Appendix A).

### 2.1. Population Structure, Kinship, and LD Analysis

Principal component analysis (PCA) was performed using high-density SNP data to explore the population structure of the experimental material. The first two principal components explained only 7.60 and 2.75 percent of the total genetic variation (Figure 2), indicating that the genetic diversity is distributed across many dimensions. Due to the low proportions of variation captured, the PCA biplot did not provide a clear separation of genotypes into distinct subpopulations. Thus, PCA was considered supplementary, and population structure was primarily assessed using the kinship matrix.

The genetic relatedness among the genotypes is presented as a kinship matrix heatmap (Figure 3), which reveals the varying degree of relatedness among the inbred lines considered. While most of the genotype pairs displayed low to moderate kinship coefficients, a few clusters exhibited higher relatedness, indicating the existence of a potential substructure in the study panel.

The genome-wide linkage disequilibrium (LD) decay analysis revealed that the linkage disequilibrium (r^2^) decreased below the critical threshold of 0.2, at a distance of approximately 300 kb. This indicates that LD extends over a relatively longer distance, suggesting moderate to low recombination in the population. The r^2^ did not drop below 0.1 within the examined range, highlighting the persistence of LD even at a larger distance (Figure 4).

### 2.2. GWAS Analysis and Candidate Gene Identification

GWAS analysis was performed for NCLB response of the experimental material across two seasons, employing both general and mixed linear models using 289,701 high-quality and dense SNP markers. The results from the pooled GWAS analysis indicated the significant association of 70 and 4 QTNs in GLM and FarmCPU models, respectively. The QTNs associated with NCLB resistance, their chromosome position, MAF, −log_10_ (*p*) value, and PVE (%) are represented in Appendix A. The identified NCLB resistance-linked QTNs were located on chromosomes 1, 2, 3, 4, 5, 6, 7, 8, and 9, with −log_10_P values ranging between 6.76 and 10.36. Furthermore, the highest variation explained by the linked QTN identified from GLM model was 12.12 percent by *S1_7356398*, located on chromosome 1. Another significantly associated QTN, *S2_51098833,* identified from the FarmCPU located on chromosome 2, explained 6.10 percent of the variation to NCLB resistance. The Manhattan plots showing the association of significant QTNs across all the chromosomes in both the GWAS models are given in Figure 5. The *p*-value of the associated QTNs displayed significant deviation from the expected *p*-value in the QQ-plot (Figure 6). 

Furthermore, the candidate genes were identified utilizing the B73 version 5 maize reference genome (https://maizegdb.org/genome/assembly/Zm-B73-REFERENCE-NAM-5.0 (accessed on 25 August 2025)). The 20 Kb upstream and downstream regions, from the physical position of the significant QTNs identified from the GWAS analysis, were scanned for the presence of key candidate genes involved in plant defence mechanisms. The 20 Kb downstream of the QTN *S1_7356398* located on chromosome 1, explaining 12.12 percent of the phenotypic variation identified from the GLM model, had two candidate genes, viz., *Zm00001eb002620* (chitinase) and *Zm00001eb002610* (putative serine/threonine protein kinase). The gene chitinase is involved in the defence response to the fungal attack, and is also involved in the polysaccharide catabolic process, cell wall macromolecule catabolic process, chitinase activity, chitin catabolic process, and protein phosphorylation. Another candidate gene, *Zm00001eb002610* (putative serine/threonine protein kinase), was found to be associated with the brassinosteroid-mediated signalling pathway, which plays a prime role in the plant’s response to several biotic and abiotic stresses. It is also involved in protein kinase activity and ATP binding. Another significantly associated QTN *S1_290805849*, located on chromosome 1, identified from the FarmCPU model, identified the candidate gene *Zm00001eb059630* (aldehyde oxygenase) in the upstream region. It is involved in fatty acid biosynthesis and metabolism, and iron ion binding. In the downstream of the identified QTN, we found another annotated gene encoding Obg-like ATPase 1; this OBG-type G domain-containing protein is involved in ATP hydrolysis, and ATP and GTP binding. The list of all the candidate genes identified by keeping B73 as a reference genome is given in Table 3.

## 3. Discussion

Breeding for biotic and abiotic stress tolerance is crucial in order to achieve climate resilience in the era of a changing climate. Northern corn leaf blight is one of the major diseases affecting maize-growing regions. Identifying sources of disease resistance and markers associated with NCLB resistance is essential for enhancing maize productivity in the current production scenario. Thus, the identification, validation, and deployment of a high-value genomic region for the target trait will accelerate the development and selection of improved maize varieties or hybrids. The genetic mapping and molecular characterization of the genomic regions associated with the trait can be achieved through targeted molecular breeding. The association mapping or the GWAS analysis exploits historical recombination to elucidate the marker–trait associations.

The ANOVA results indicated substantial genetic variation in the NCLB disease response across years, suggesting the potential for selecting resistant genotypes. The majority of the experimental material displayed a moderately resistant response, whereas only a few were completely resistant, highlighting the skew of the genotypes towards susceptible reaction. Significant genotype × year interactions underscored the influence of environmental factors on the disease response. The higher magnitude of genetic variability parameters, viz., PCV, GCV, heritability, and genetic advance as percent mean indicated the predominance of additive gene action in disease response, implying the effectiveness of selection. Quantitative inheritance of NCLB resistance with additive gene action was reported earlier by [6,7,22].

Furthermore, the population structure and relatedness among the inbreds were evaluated using PCA and kinship analysis to ensure reliable association mapping. The relatively low proportion of variation explained by each of the principal components indicated the minimal confounding due to population structure, thereby reducing the likelihood of spurious marker–trait associations [23]. To overcome this, advanced mixed models such as MLM, CMLM, MLMM, FarmCPU, BLINK, and SUPER, which incorporate both population structure and kinship information, provide more robust association detection [15]. In the present study, the application of both general and mixed linear (single and multi-locus) models ensured the effective control of type I errors and enhanced the detection power. Moreover, the observed LD pattern suggested that the SNP density used was adequate for achieving the resolution needed for the reliable QTN detection across the genome. Overall, the diversity, moderate relatedness, and appropriate marker density in the current study validates its suitability for GWAS analysis in maize.

GWAS identified significant associations with NCLB resistance from GLM and FarmCPU models. It was noted that the multi-locus GWAS models, like FarmCPU, outperform the single-locus models(GLM). The single-locus models were relatively successful for the traits with high heritability. The extent of identification of true positives is high for single-locus models for the traits with high heritability; however, the number of identified false positives also increases roughly at the same time. This issue becomes particularly evident with the FDR correction of *p*-values at the 5 percent significance level. Among the various multi-locus models, the FarmCPU is preferred because it gives fewer false positives with FDR at a 5% significance level [24,25,26,27,28] compared the efficiency of several GWAS models for various traits in different crops and pointed out that the multi-locus models, viz., MLMM and FarmCPU, were more efficient in identifying associations.

In addition to this, the QTNs identified from the current study were found to be located in annotated genes with functional domains implicated in defence mechanisms in maize and other crops. The highly significant QTN *S1_7356398* on chromosome 1, identified from the GLM model, had two candidate genes, viz., *Zm00001eb002620* (chitinase) and *Zm00001eb002610* (putative serine/threonine protein kinase), within its downstream region. Chitinases are a group of pathogenesis-related (PR) proteins that hydrolyze chitin, a key structural component of fungal cell walls, thereby restricting the pathogen invasion and spread. Enhanced expression of the gene has been linked to resistance against several fungi. Cazares-Alvarez et al. [21] found that the chitinase family genes are involved in plant development, hormone response, and abiotic stress response, along with defence response against infection by *Fusarium verticillioides* (Sacc.) Nirenberg, which causes Fusarium stalk rot in maize.

The nearby *Zm00001eb002610* gene encodes a putative serine/threonine protein kinase, a class of signalling proteins that are involved in phosphorylation cascades in plant immune responses. Serine/threonine kinases have been reported to participate in brassinosteroid-mediated signalling pathways that regulate both biotic and abiotic stress tolerance [29,30].

Another significantly associated QTN, *S1_290805849,* also located on chromosome 1 and identified by the FarmCPU model, was found near the *Zm00001eb059630* (aldehyde oxygenase) gene. This gene is primarily involved in fatty acid biosynthesis and metabolism, processes known to influence membrane stability and defence signalling under pathogen attack. In sorghum, aldehyde oxygenase (SORBI_3004G218100) was differentially expressed in grains infected with the smut fungus *Sporisorium reilianum*, suggesting its conserved role in plant defence against fungus [31]. Collectively, these candidate genes highlight potential biochemical and signalling pathways that may contribute to NCLB resistance in maize through strengthened cell wall defence, hormone signalling, and oxidative metabolism.

Although we did not perform expression or functional assays in the current study, the QTNs identified in proximity to *Zm00001eb002620, Zm00001eb002610*, and *Zm00001eb059630* are biologically plausible candidates for NCLB resistance. Their established roles in pathogen recognition, defence signalling, and secondary metabolite production in maize and related species [21,29,31], together with the statistical association with resistance phenotypes, provide a strong rationale for further experimental validation. Future studies, including transcript profiling and functional characterization, are needed to confirm their roles in mediating NCLB tolerance.

## 4. Materials and Methods

The experimental material consisted of 336 maize inbred lines procured from CYMMIT, Hyderabad, India. These lines were evaluated for their response to NCLB disease under artificial epiphytotic conditions in Hassan for two seasons (rainy seasons of 2023 and 2024) in an alpha lattice design with two replications. Each inbred was sown in a 2 m row with 60 cm between the rows and 20 cm between the plants, ensuring uniform plant density and population equilibrium across the field. Standard agronomic and plant protection measures were followed, except for the targeted foliar disease, to promote natural infection.

### 4.1. Screening for NCLB

A pure culture of *Exserohilum turcicum* was obtained and grown on the PDA media. It was used for inoculating the sterile sorghum grains. The inoculated sorghum grains were then cultured at room temperature for two weeks. The colonized grains were ground into a fine powder, which was used for inoculating the test entries 30 days after planting. A pinch of grounded powder was applied to each plant’s whorl, followed by a water spray. Observations on disease symptoms were recorded 75 days after planting, following a 1–9 scale given by [32,33]. Plants with a disease score ≤3 were categorized as resistant, 3.1 to 5 as moderately resistant, 5.1 to 7 as moderately susceptible, and above 7 as susceptible. The phenotype of plants for different disease scores is given in Figure 7.

### 4.2. Statistical Analyses

A mixed linear model was employed to analyze phenotypic data from the alpha lattice design, considering genotypes, environments, and the interactions between genotype and environment, as well as replication and environment, as random effects.$$Y_{ijko}=\mu +g_{i}+l_{j}+r_{kj}+b_{ojk}+e_{ijko}$$
where $Y_{ijko}$ is the phenotypic performance of the ith genotype at the jth environment in the kth replication of the oth incomplete block, μ is the intercept term, $g_{i}$ is the genetic effect of the ith genotype, $l_{j}\;$is the effect of the jth environment, $r_{kj}$ is the effect of the kth replication at the jth environment, $b_{ojk}$ is the effect of the oth incomplete block in the kth replication at the jth environment, and $e_{ijko}$ is the residual.

The BLUPs were computed using Meta R version 8.1, using the following general linear model:$$Y_{ijkl}=\mu +{Env}_{i}+g_{j}+R_{k}\left({Env}_{i}\right)+Block(R_{k}\left({Env}_{i}\right)){+e}_{ijkl}$$

$Y_{ijkl}$ = NCLB response of each genotype.

μ = Overall mean.

${Env}_{i}$ = Fixed effect of the ith environment.

$g_{j}$∼N (0, $\sigma_{g}^{2}$) = Random effect of the jth genotype.

$R_{k}({Env}_{i})$ ∼N (0, $\sigma_{R}^{2}$) = Random effect of the kth replication nested within environment.

$Block\;(R_{k}{Env}_{i})∼N(0,\;\sigma_{B}^{2})$ = Random effect of the lth block nested within replication × environment.

$e_{ijkl}∼N(0,\;\sigma_{e}^{2})$ = Residual error.

Furthermore, the genetic variability parameters viz., phenotypic coefficient of variation (PCV), genotypic coefficient of variation (GCV), broad-sense heritability, and genetic advance as percent mean are computed.

The formula for the calculation of PCV and GCV are given below [34]:$$\mathrm{PCV}\;(\%)\;=\;\frac{\sqrt{\sigma_{p}^{2}}}{\bar{x}}\;\times \;100$$
where

$\sigma_{p}^{2}$—phenotypic variance, and $\bar{X}$—overall mean;$$\mathrm{GCV}\;(\%)=\frac{\sqrt{\sigma_{g}^{2}}}{\bar{x}}\;\times \;100$$
where

$\sigma_{g}^{2}$—genotypic variance, and $\bar{X}$—overall mean.

The expected genetic advance as a percent of the mean was estimated as follows [35] $$\mathrm{GAM}=\frac{GA}{\mu }\;\times \;100$$
where GA is the genetic advance and μ is the general mean.

Genetic advance is computed using the formula given below:$$GA\;=\;k\;\times \;h_{b}^{2}\;\times \sqrt{\sigma_{p}^{2}}$$
where k = selection differential (2.06) at 5% selection intensity, and

$\surd \sigma_{p}^{2}$ = phenotypic standard deviation,

$h_{b}^{2}$ = broad senese heritability.

The broad-sense heritability across multi-season data was estimated as follows:$$H^{2}=\frac{\sigma_{g}^{2}}{{(\sigma }_{g\;}^{2}+\frac{\sigma_{ge}^{2}}{e}+\;\frac{\sigma_{e}^{2}}{er})}$$
where $\sigma_{g}^{2}$, $\sigma_{ge}^{2}$ and $\sigma_{e}^{2}$ are the genotypic, genotype by environment interactions, and error variance components, respectively; and r and e are the number of replications and environments within each of the environments included in the analysis.

### 4.3. Genotyping

The 336 maize inbreds from CYMMIT were genotyped using GBS, and data concerning 955,690 SNPs was obtained. The marker data was filtered for percent heterozygosity (0.25), minor allele frequency (0.05), and call rate (0.70) using TASSEL version 5.2 [36]. A total of 289,701 SNPs retained were used in further analysis.

### 4.4. Population Structure and LD Analysis

The population structure existing in the experimental material was assessed utilizing principal component analysis. The genome-wide LD between all possible pairs of SNPs was estimated using the weighted average of squared allele frequency correlations (r^2^), using the ‘gapit’ package of R software version 4.4.3. The LD decay curve was obtained by analyzing the LD between all possible pairs of SNPs to visualize genome-wide LD patterns.

### 4.5. GWAS Analysis

GWAS analysis was performed using six different models that include both single- and multi-locus models. The general linear model (GLM) is a basic fixed-effects single-locus model that incorporates population structure through principal components (PCs) or a Q-matrix; however, it lacks a kinship component, making it highly prone to false positives [37]. The mixed linear model (MLM), another single-locus model, overcomes this limitation by including both population structure (Q or PCs) and a kinship matrix (K), thus controlling for both relatedness and structure [38]. To enhance computational efficiency, the compressed MLM (CMLM) model clusters individuals based on genetic similarity before fitting the MLM [39]. The multi-locus mixed model (MLMM) further refines this approach, by incorporating significant marker cofactors identified through stepwise regression, increasing its power to detect multiple loci [40]. FarmCPU (Fixed And Random Model Circulating Probability Unification) iteratively separates fixed and random effects, using PCs and selected markers as covariates to avoid overfitting and to reduce false positives [28]. BLINK (Bayesian Information and Linkage Disequilibrium Iteratively Nested Keyway) improves upon FarmCPU by removing the kinship matrix altogether and utilizing LD information, thereby enhancing computational speed and power, especially for large datasets. Each model applies a unique combination of covariates, such as Q-matrices, PCs, kinship matrices, and marker cofactors to effectively control population structure, genetic relatedness, and marker confounding, and is selected based on the genetic architecture of the trait and computational resources available. All six GWAS models were employed to identify the marker associations linked to NCLB disease resistance.

### 4.6. QQ-Plot and Manhattan Plot

The QQ-plot is the graphical representation of the deviations of the observed *p*-values from the normal distribution. The observed *p*-values from each of the SNP markers are plotted against expected values from a theoretical χ^2^-distribution to study the genomic inflation using R software’s ‘gapit’ package version 4.4.3. Manhattan plots were plotted, with the genomic coordinates displayed along the X-axis, and the negative logarithm of the association *p*-value for each SNP marker displayed on the Y-axis, indicating the significant association of SNP markers to disease resistance.

## 5. Conclusions

This study demonstrates the effectiveness of genome-wide association studies in dissecting the genetic basis of northern corn leaf blight (NCLB) resistance in maize. The QTNs identified in proximity to genes such as chitinase, serine/threonine protein kinase, and aldehyde oxygenase suggest key roles for pathogen recognition, defence signalling, and secondary metabolite production in mediating resistance. These insights enhance our understanding of the molecular mechanisms underlying quantitative resistance to NCLB and provide valuable targets for the marker-assisted selection and breeding of resistant maize cultivars. Future studies incorporating expression profiling and the functional validation of these candidate genes will be crucial for translating these associations into durable resistance strategies in maize production.

## Figures and Tables

**Figure 1 ijms-26-10677-f001:**
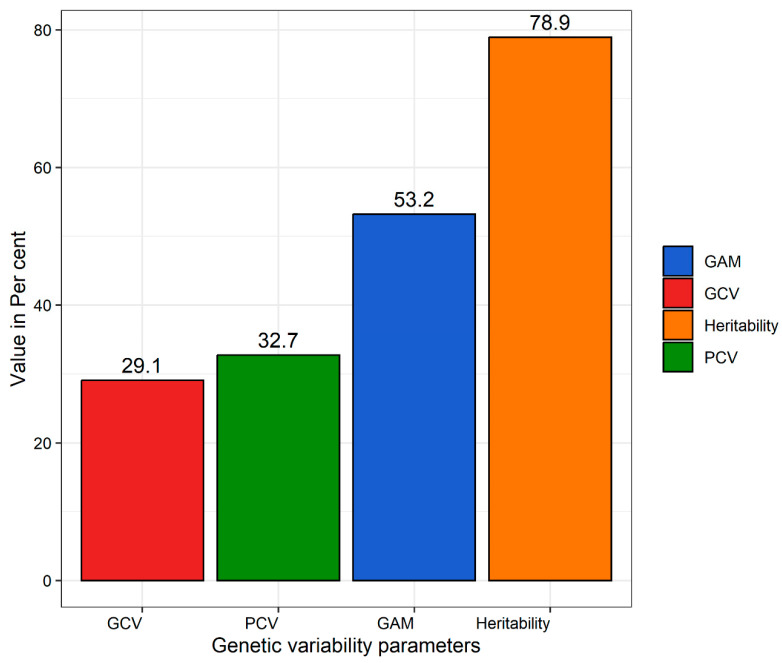
Estimates of genetic variability parameters for northern corn leaf blight (NCLB) disease response in 336 maize inbred lines evaluated across two growing seasons. Parameters include genotypic coefficient of variation (GCV), phenotypic coefficient of variation (PCV), genetic advance as percent of mean (GAM), and broad-sense heritability (H^2^). GCV and PCV were calculated following standard quantitative genetic methods, while GAM represents the expected improvement in the trait mean under selection (refer to material and methods).

**Figure 2 ijms-26-10677-f002:**
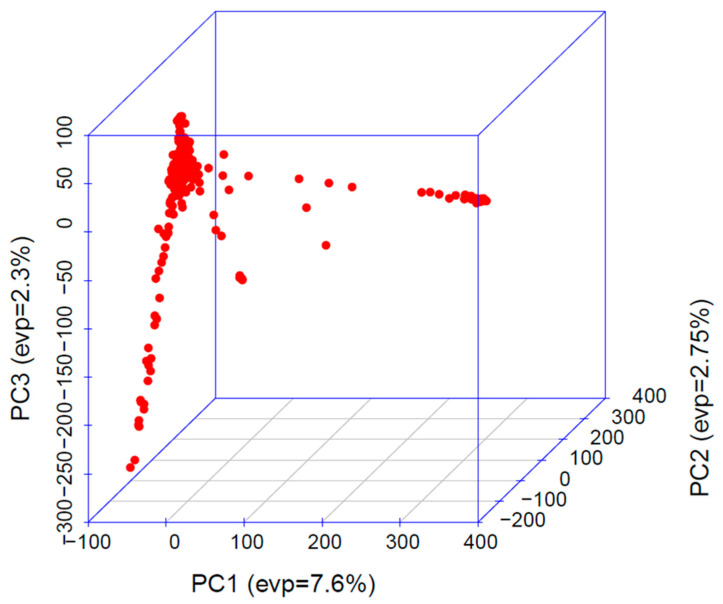
Three-dimensional PCA plot showing the population structure of the experimental material. PC1, PC2, and PC3 explain 7.6%, 2.75%, and 2.3% of the total variation, respectively (Red colour dots indicate the maize inbreds grouping into three different clusters). Clustering of points indicates genetic differentiation among the individuals.

**Figure 3 ijms-26-10677-f003:**
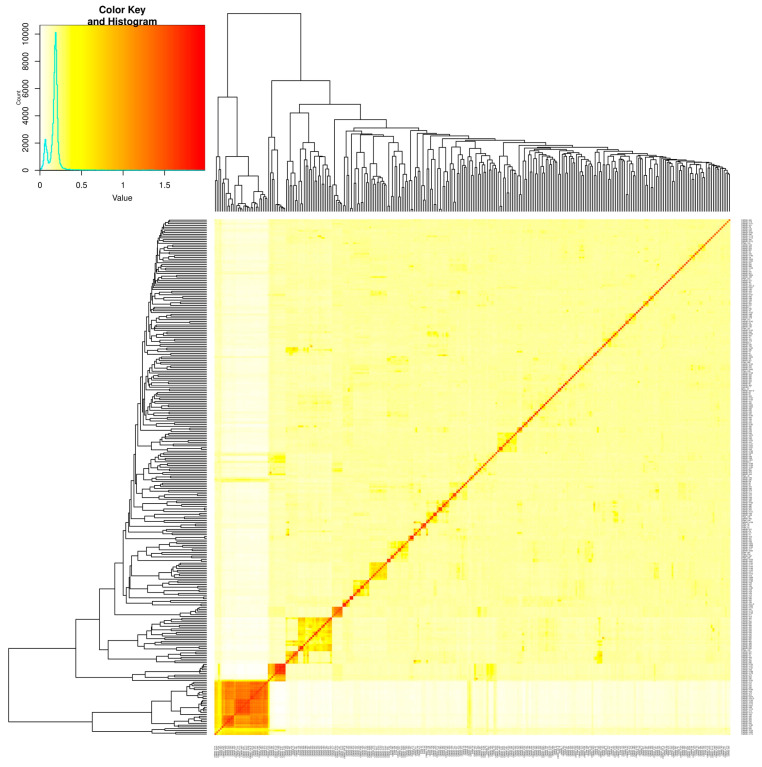
VanRaden’s algorithm-based kinship plot representing the genetic relatedness among 336 maize inbreds (increasing color intensity from yellow to brick red indicates high relatedness).

**Figure 4 ijms-26-10677-f004:**
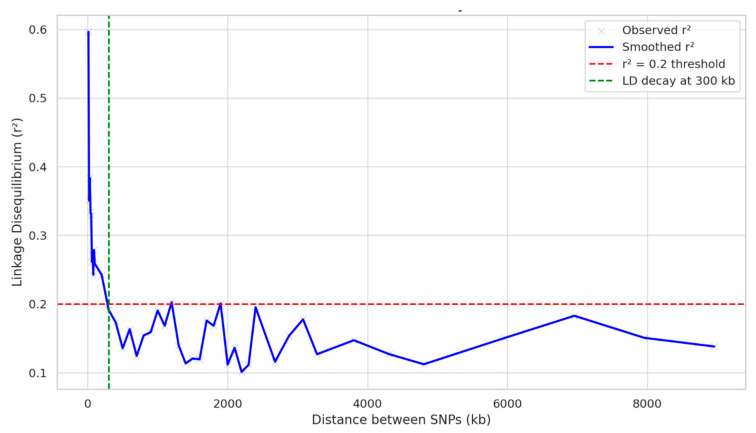
LD decay pattern across the genome at r^2^ threshold of 0.2.

**Figure 5 ijms-26-10677-f005:**
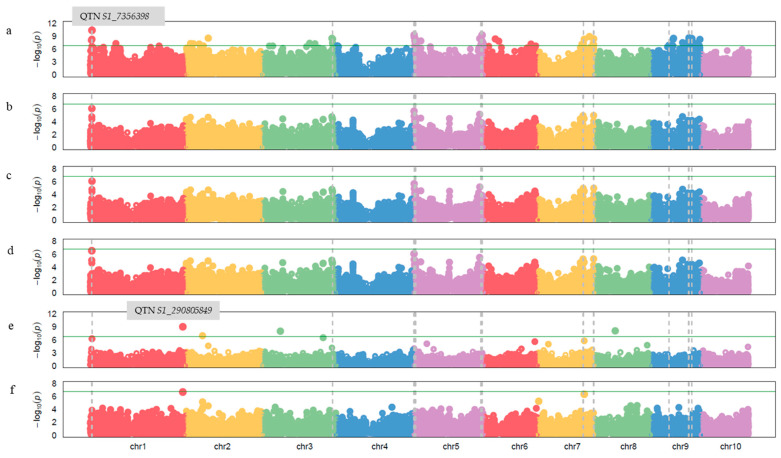
Manhattan plots for depicting marker-trait associations for northern corn leaf blight disease resistance across different GWAS models ((**a**) GLM, (**b**) MLM, (**c**) CMLM, (**d**) MLMM, (**e**) FarmCPU, and (**f**) BLINK). The QTNs located above the threshold −log10(*p*) are considered to be significantly associated with NCLB resistance. Different colours indicate different maize chromosomes.

**Figure 6 ijms-26-10677-f006:**
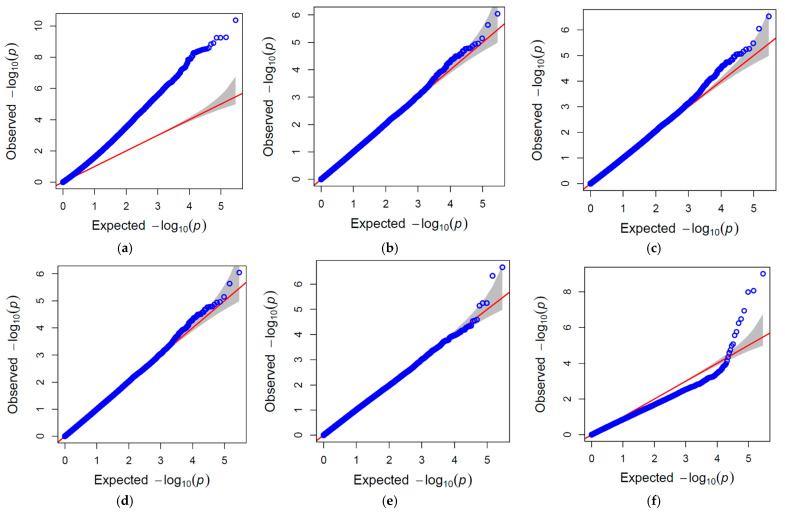
Quantile–quantile (QQ)-plots (graphical representations of the deviations of the observed *p*-values from the normal distribution. The observed *p*-values from each of the SNP markers are plotted against the expected values from a theoretical χ^2^-distribution to study the genomic inflation) depicting the distribution of the observed versus expected *p*-values for marker–trait associations for NCLB resistance under different GWAS models, including (**a**) GLM, (**b**) MLM, (**c**) MLMM, (**d**) CMLM, (**e**) BLINK, and (**f**) FarmCPU, indicating the control of false positives across models. Blue circles represent observed p-values, the red line indicates the expected distribution under the null hypothesis, and the gray shaded area corresponds to the 95% confidence interval. Points above the red line with significant deviation suggest potential significant marker-trait associations.

**Figure 7 ijms-26-10677-f007:**
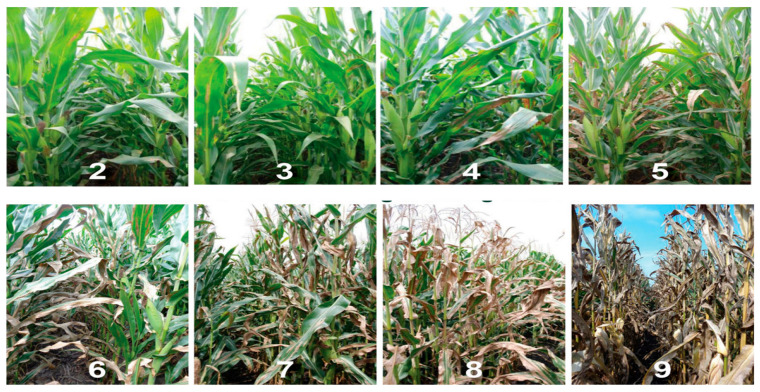
Figure depicting disease scoring scale for northern corn leaf blight response.

**Table 1 ijms-26-10677-t001:** Analysis of variance for northern corn leaf blight disease response of maize inbreds in Hassan for two seasons.

Source of Variation	Degrees of Freedom	Mean Sum of Squares	
		Season 1	Season 2
Entry	335	2.06 ***	2.42 ***
Replication	1	1.74	1.90 *
Replication: Block	22	1.66	1.71
Residuals	313	0.24	0.77

***, and * indicate significance at 0.0001 and 0.05 percent probability, respectively.

**Table 2 ijms-26-10677-t002:** Pooled ANOVA for NCLB disease response across seasons.

Source of Variation	Degrees of Freedom	Mean Sum of Squares
Genotype	335	117.70 ***
Year	1	7.70 *
Replication	1	6.46
Genotype: Year	335	48.59 **
Residuals	1339	1.40

***, ** and * indicate the significance at 0.0001, 0.001 and 0.05 percent probability, respectively.

**Table 3 ijms-26-10677-t003:** List of putative candidate genes identified from the current study.

QTNs	Chromosome	Candidate Gene	Gene Name	Predicted Function	Reference
S1_7356398 (downstream)	1	*GRMZM2G099598*	Chitinase	Defence response to fungus Polysaccharide catabolic process Cell wall macromolecule catabolic process Chitinase activity Chitin catabolic process	[21]
	1	*GRMZM2G099598*	Putative serine/threonine protein kinase	Protein phosphorylation Brassinosteroid mediated signalling pathway Protein kinase activity ATP binding Salt and drought tolerance	-
S1_290805849 (upstream)	1	*GRMZM2G038964*	Aldehyde oxygenase (deformylating)	Fatty acid biosynthetic process Iron ion binding Fatty acid alpha-hydroxylase activity Aldehyde oxygenase (deformylating) activity Fatty acid metabolic process Octadecanal decarbonylase activity	-
S1_290805849 (downstream)	1	*GRMZM2G038494*	Obg-like ATPase 1; OBG-type G domain-containing protein	ATP hydrolysis activity Ribosomal large subunit binding ATP binding GTP binding	-

## Data Availability

The original contributions presented in this study are included in the article/Appendix A. Further inquiries can be directed to the corresponding authors.

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
