# Peer review of "Genome-Wide Association Study Reveals Novel QTNs and Candidate Genes Implicated in Resistance to Northern Corn Leaf Blight in Maize (Zea mays L.)"

_ijms, 2025, doi:10.3390/ijms262110677_

Round 1

Reviewer 1 Report

Comments and Suggestions for Authors

The study focuses on resistance to Northern Corn Leaf Blight (NCLB), a disease caused by the fungus Exserohilum turcicum, which affects maize production worldwide. A genome-wide association study (GWAS) was conducted on 336 inbred maize lines to identify genetic markers (QTNs) and candidate genes associated with resistance to this disease.

While the study is interesting, several corrections are required. For instance, the objective of the work should be clearly stated, the relevance of the study should be highlighted, the validity of the applied analyses—such as the principal component analysis (PCA)—should be verified, and some minor corrections should be made.

The following observations are noted:

The abbreviation GWAS should be written out in full the first time it appears in the abstract (line 19).

In line 39, when mentioning Northern Corn Leaf Blight, the abbreviation NCLB should be included in parentheses.

Lines 42 and 43 should be corrected: the currently valid name of the species is Setosphaeria turcica, and Exserohilum turcicum is considered its anamorphic stage; however, both refer to the same species.

In the introduction, write out the abbreviation GWAS in full the first time it appears (line 87).

The objective of the study should be clearly stated at the end of the introduction, and it should be specified that 336 maize varieties will be used—or at least mention that a diverse panel of maize lines will be used to identify candidate genes.

Remove the extra space between lines 117 and 118.

Replace the word viz with a colon ( : ) in line 118.

The abbreviations PCV and GCV in line 120 should be written out in full.

The figure legend for Figure 1 should include all abbreviations followed by their definitions, then a proper description of the figure.

Since the PCA explains only 13% of the population variation, it is not appropriate to include it as evidence of population structure. However, it can be mentioned that PCA was used to generate the analysis shown in Figure 3.

Figure 2, which shows the PCA, should not be included since the distribution of the maize varieties is not valid due to the low percentage of variation explained by each principal component.

The abbreviation LD in line 151 should be written out in full the first time it appears.

For Figure 5, it is recommended to label the key peaks with the names of the associated genes and to clearly explain in the figure legend what is being evaluated in this analysis.

Throughout the text, use Anglo-American number formatting. For example, instead of writing 2,89,701, write 289,701 (line 161).

Remove “the” before GWAS models in line 171.

The statement between lines 251–252 is not valid for the PCA analysis, due to the low percentage of explained variation. Here, the authors may instead discuss that PCA was not useful for visualizing population structure, but was used to reconstruct the heatmap shown in Figure 3.

In the Discussion, include a final paragraph outlining the significance of the findings and how the information generated could be improved or used in the future, as it is currently unclear how the identification of these genes would be applied.

Author Response

Comment 1: The abbreviation GWAS should be written out in full the first time it appears in the abstract (line 19).

Response: Abbreviation for the GWAS is added as suggested by the learned reviewer.

Comment 2: In line 39, when mentioning Northern Corn Leaf Blight, the abbreviation NCLB should be included in parentheses.

Response: Abbreviation of NCLB is added as suggested by the learned reviewer.

Comment 3: Lines 42 and 43 should be corrected: the currently valid name of the species is Setosphaeria turcica, and Exserohilum turcicum is considered its anamorphic stage; however, both refer to the same species.

Response: Corrected as suggested by the learned reviewer.

Comment 4: In the introduction, write out the abbreviation GWAS in full the first time it appears (line 87).

Response: Abbreviation for the term GWAS is added as suggested by the learned reviewer.

Comment 5: The objective of the study should be clearly stated at the end of the introduction, and it should be specified that 336 maize varieties will be used—or at least mention that a diverse panel of maize lines will be used to identify candidate genes.

Response: The objective of the study is clearly stated at the end of the introduction as suggested by the learned reviewer.

Comment 6: Remove the extra space between lines 117 and 118.

Response: The extra space between lines 117 and 118 was removed as suggested by the reviewer.

Comment 7: Replace the word viz with a colon ( : ) in line 118.

Response: The word viz., was removed and a colon was added as suggested by the reviewer (lines 131-132).  

Comment 8: The abbreviations PCV and GCV in line 120 should be written out in full.

Response: The abbreviations for PCV and GCV were added in lines 133-134 as suggested by the learned reviewer.

Comment 9: The figure legend for Figure 1 should include all abbreviations followed by their definitions, then a proper description of the figure.

Response: Figure 1 includes an abbreviation for all the parameters and the description of the figure is added as suggested by the reviewer. Details on the calculation of genetic variability parameters are added in material and methods as suggested by the learned reviewer.

Comment 10: Since the PCA explains only 13% of the population variation, it is not appropriate to include it as evidence of population structure. However, it can be mentioned that PCA was used to generate the analysis shown in Figure 3.

Response: The PCA results and their inference are modified as suggested by the learned reviewer.

Comment 11: Figure 2, which shows the PCA, should not be included since the distribution of the maize varieties is not valid due to the low percentage of variation explained by each principal component.

Response: We acknowledge that the variation explained by the first three principal components is relatively low, such values are common in GWAS studies involving diverse maize germplasm due to its high genetic complexity. The PCA was primarily used to estimate population structure for inclusion as covariates in the GWAS models, rather than to describe major population groupings. Therefore, we have retained the PCA figure to illustrate this aspect of the analysis but, we have clarified its purpose and interpretation in the revised manuscript.

Comment 12: The abbreviation LD in line 151 should be written out in full the first time it appears.

Response: The abbreviation for LD is added in line 155 as suggested by the learned reviewer.

Comment 13: For Figure 5, it is recommended to label the key peaks with the names of the associated genes and to clearly explain in the figure legend what is being evaluated in this analysis.

Response: Figure 5 is modified as suggested by the learned reviewer.

Comment 14: Throughout the text, use Anglo-American number formatting. For example, instead of writing 2,89,701, write 289,701 (line 161).

Response: Anglo-American number formatting is followed as suggested by the learned reviewer throughout the manuscript.

Comment 15: Remove “the” before GWAS models in line 171.

Response: The word was removed as suggested by the reviewer.

Comment 16: The statement between lines 251–252 is not valid for the PCA analysis, due to the low percentage of explained variation. Here, the authors may instead discuss that PCA was not useful for visualizing population structure, but was used to reconstruct the heatmap shown in Figure 3.

Response: The information about PCA in the lines 251-252 was modified and inference for low per cent variation explained by each of the PC was added as suggested by the learned reviewer from lines 246- 252.

Comment 17: In the Discussion, include a final paragraph outlining the significance of the findings and how the information generated could be improved or used in the future, as it is currently unclear how the identification of these genes would be applied.

Response: A paragraph outlining the significance of findings and future line of work was added as suggested by the learned reviewer.

Reviewer 2 Report

Comments and Suggestions for Authors

The article "Genome-wide association study reveals novel QTNs and candidate genes implicated in resistance to Northern corn leaf blight in maize (Zea mays L.)" by M. Udaya Shetty examines the identification of potential markers for resistance to late blight.
The manuscript is well-written and contains the necessary sections; however, both its structure and formatting preclude its acceptance in its current form.
The study clearly has a practical context, but I found no evidence supporting the assumptions through any experiments, even laboratory ones. Furthermore, the lack of hypotheses, methodology other than bioinformatics analysis, and presentation of results in the introduction, where the theoretical basis, problems, and possible solutions, as well as potential applications, should have been, is critically unclear. The results are also presented uniformly in the results, discussion, and conclusion sections. This somewhat obscures the study's potential meaning. The initial problem with the paper is that the authors somehow assume that late blight is a fungus. It is now known that oomycetes, which have significant differences in both their life cycle and their strategy for interacting with plants, hardly warrant this designation.
The authors then list a number of candidate genes that, in their view, may confer resistance, without discussing in detail how their presence or putative expression advances the question of genotype assessment. However, such data should be presented and discussed for each proposed gene or group of genes; otherwise, I fundamentally fail to understand the point of such speculative analyses.
I recommend that the authors first understand the taxonomy, then remove the results from the introduction, outline specific objectives, and, in the conclusion, express a judgment on whether they have achieved a result or not. This structure creates the impression that the authors have discovered that these genes work, which is not supported by any evidence, and are simply describing that they have observed their presence, which is also highly arbitrary. I would like to see a clear structure and consistency in the presentation.
In addition to repeating the presentation, the discussion needs to clarify whether this technology can be useful and how exactly the link between the mentioned genes and resistance should be further identified.
Please remove the words "fungal pathogens" and "genetic gain" from the text, as these should either be defined or explained as a term (which is unlikely).
Please clarify what you mean by "genetic advance as a percentage mean and broad sense of her-119 resistance were estimated (Figure 1)." You should be specific and adhere to the terminology, as understanding what you are writing is quite problematic.
The article's illustrations and figure captions further complicate the situation. In IJMS, figures can often be viewed outside of the article; for example, they can be searched online or presented on scientific platforms. However, in this manuscript, the captions lack any method, object, or specifics, and often it's completely impossible to understand what's where because there are no captions.
I would like to remind the authors that in interdisciplinary journals, figures and text should be useful and intuitively understandable to specialists in related fields.
Another problem is that in agronomy, a minimum range of three years is considered appropriate, and if two years are cited, especially for such a complex disease, this should be substantiated.
It is also necessary to provide photographs confirming the phenotyping work, detail the evaluation methods, and clearly indicate the number of replicates, population equilibration, and statistical significance.
In this case, correlation analysis should perhaps be included if the authors have sufficient data.
I believe the authors should carefully revise the manuscript before submission.

Author Response

The article "Genome-wide association study reveals novel QTNs and candidate genes implicated in resistance to Northern corn leaf blight in maize (Zea mays L.)" by M. Udaya Shetty examines the identification of potential markers for resistance to late blight.
The manuscript is well-written and contains the necessary sections; however, both its structure and formatting preclude its acceptance in its current form.

Comment 1: The study clearly has a practical context, but I found no evidence supporting the assumptions through any experiments, even laboratory ones. Furthermore, the lack of hypotheses, methodology other than bioinformatics analysis, and presentation of results in the introduction, where the theoretical basis, problems, and possible solutions, as well as potential applications, should have been, is critically unclear.

Response: The introduction section is rewritten, including the theoretical basis of GWAS and its potential applications as suggested by the learned reviewer. Further, the results, including numeric values, were removed in the introduction section. Also, the hypothesis was stated at the end of the introduction.

Comment 2: The results are also presented uniformly in the results, discussion, and conclusion sections. This somewhat obscures the study's potential meaning.

Response: The discussion and conclusions sections have been thoroughly revised as per the suggestions of the esteemed reviewer.

Comment 3: The initial problem with the paper is that the authors somehow assume that late blight is a fungus. It is now known that oomycetes, which have significant differences in both their life cycle and their strategy for interacting with plants, hardly warrant this designation.
Response:
We would like to clarify that our study exclusively involved Exserohilum turcicum, the fungus responsible for northern corn leaf blight in maize. It was first reported to be caused by Helminthosporium turcicum Pass. in Italy by Passerini (1876). Later, Pammel et al. (1910) and Drechsler (1923) renamed the pathogen as Trichometasphaerica turcica Luttrell (Luttrell, 1958). Later, Leonard and Suggs in 1974 renamed the causal organism as Setospharerica turcica and the conidial stage as Exserohilum turcicum (Leonard and Suggs, 1974). Further, several research findings also support that northern corn leaf blight is a fungal disease caused by actinomycete fungus (Setospharerica turcica and the conidial stage as Exserohilum turcicum) (Ijaz and Fan, 2024; Gu et al., 2025; Wu et al., 2025).

In the present study, a pure culture of the fungus was isolated and maintained on potato dextrose agar (PDA). Sterile sorghum grains were subsequently inoculated with the fungal culture to produce the inoculum, which was applied to maize plants 30 days after planting. Disease assessment was carried out 75 days after planting, following the procedure detailed in the Materials and Methods section (lines 339–347). No oomycete pathogens were involved in this study.

Taxonomy of the fungus is as follows,

 Kingdom: Fungi

 Phylum: Ascomycota

 Class: Dothideomycetes

 Order: Pleosporales

 Family: Pleosporaceae

 Genus: Exserohilum

 Species: Exserohilum turcicum

Comment 4: The authors then list a number of candidate genes that, in their view, may confer resistance, without discussing in detail how their presence or putative expression advances the question of genotype assessment. However, such data should be presented and discussed for each proposed gene or group of genes; otherwise, I fundamentally fail to understand the point of such speculative analyses.

Response:
The candidate gene discussion section has been revised to provide a more detailed interpretation of each gene’s potential role in NCLB resistance. For each identified QTN, we now describe the biological functions of the proximal candidate genes—such as chitinase, serine/threonine-protein kinase, and aldehyde oxygenase—and discuss how these functions are plausibly linked to pathogen recognition, defence signalling, and secondary metabolite-mediated responses. Although expression or functional validation assays were not conducted in this study, the strong statistical associations of these loci with resistance phenotypes, together with previously reported functional evidence from maize and related species, support their potential involvement in NCLB resistance. We further emphasize that future transcriptomic and functional analyses will be required to confirm these roles.

Comment 5: I recommend that the authors first understand the taxonomy, then remove the results from the introduction, outline specific objectives, and, in the conclusion, express a judgment on whether they have achieved a result or not. This structure creates the impression that the authors have discovered that these genes work, which is not supported by any evidence, and are simply describing that they have observed their presence, which is also highly arbitrary. I would like to see a clear structure and consistency in the presentation.

Response: In accordance with the reviewer’s comments, we have revised the manuscript to enhance its overall structure and clarity. Results that were previously included in the Introduction have been removed, ensuring that this section now focuses exclusively on the background and rationale of the study. The specific objectives have been explicitly stated for greater clarity. The Conclusion has been rewritten to accurately reflect the actual outcomes of the research—specifically, the identification of QTNs associated with NCLB resistance and the recognition of biologically plausible candidate genes, without implying functional validation. These revisions collectively improve the coherence, organization, and presentation of the manuscript.
Comment 6: In addition to repeating the presentation, the discussion needs to clarify whether this technology can be useful and how exactly the link between the mentioned genes and resistance should be further identified.

Response: The repetitive content in the presentation has been removed, and the Discussion section has been rewritten to emphasize the relevance and applicability of the technology used in this study. Additionally, the methods employed to establish the association between the identified genes and resistance have been clearly described.

Comment 7: Please remove the words "fungal pathogens" and "genetic gain" from the text, as these should either be defined or explained as a term (which is unlikely).
Response:
The terms “fungal pathogens” and “genetic gain” have been removed from the text as per the reviewer’s suggestion.

Comment 8: Please clarify what you mean by "genetic advance as a percentage mean and broad sense of heritability-119 resistance were estimated (Figure 1)." You should be specific and adhere to the terminology, as understanding what you are writing is quite problematic.
The article's illustrations and figure captions further complicate the situation. In IJMS, figures can often be viewed outside of the article; for example, they can be searched online or presented on scientific platforms. However, in this manuscript, the captions lack any method, object, or specifics, and often it's completely impossible to understand what's where because there are no captions.

Response: In accordance with the reviewer’s suggestions, all figure captions have been revised for clarity and consistency.
Comment 9: I would like to remind the authors that in interdisciplinary journals, figures and text should be useful and intuitively understandable to specialists in related fields.
 Response:
Both the figure captions and the main text have been revised as suggested by the learned reviewer.

Comment 10: Another problem is that in agronomy, a minimum range of three years is considered appropriate, and if two years are cited, especially for such a complex disease, this should be substantiated.

Response: We acknowledge the reviewer’s concern regarding the standard practice of evaluating complex disease traits across at least three seasons and multiple locations. In our study, phenotypic data collected over two consecutive seasons at the same location exhibited consistent genotypic trends, indicating reliable differentiation of disease responses. The trials were conducted under uniform agro-climatic and management conditions, thereby minimizing environmental variation. We agree that including additional seasons and locations would further strengthen the robustness of the conclusions, and we plan to incorporate these aspects in future investigations.

Comment 11: It is also necessary to provide photographs confirming the phenotyping work, detail the evaluation methods, and clearly indicate the number of replicates, population equilibration, and statistical significance.

Response: Photographic evidences confirming the evaluation methods are included. Further, details on population equilibrium and statistical significance are added in the material and methods section.

Comment 12: In this case, correlation analysis should perhaps be included if the authors have sufficient data.

Response: We have evaluated the panel of inbred lines only for NCLB response and hence, correlation analysis is not required.

References

Ijaz, B. and Fan, X. 2024, Understanding Northern corn leaf blight (NCLB) disease resistance in maize: Past developments and future directions. Plant Stress, https://doi.org/10.1016/j.stress.2024.100625

Gu, Y., Yan, B., Yang, Y., Huang, Y. et al. 2025, Metabolomic Analysis of Maize Response to Northern Corn Leaf Blight. 15(2):113. doi: 10.3390/metabo15020113

Wu, J., Yang, W., Shi, X., Zhang, B., et al. 2025, Identification and fine-mapping of qNCLB3.04 resistant to Northern Corn Leaf Blight. Molecular Breeding, 45:59.

Round 2

Reviewer 1 Report

Comments and Suggestions for Authors

Upon reviewing the manuscript again, I noticed that the authors have addressed the proposed comments and also incorporated additional crucial points, which have notably improved the work. I did not find any new issues, and therefore, the manuscript can be accepted for publication in its current form.

Reviewer 2 Report

Comments and Suggestions for Authors

The article "Genome-wide association study reveals novel QTNs and candidate genes implicated in resistance to Northern corn leaf blight in maize (Zea mays L.)" by M. Udaya Shetty et al. has been significantly revised by the authors. Key comments have been addressed, and figure captions have been corrected. I believe this manuscript is suitable for publication.
However, I recommend improving the quality of Figure 3, as the captions are not identifiable even at high magnification.